# A Waveform Mapping-Based Approach for Enhancement of Trunk Borers’ Vibration Signals Using Deep Learning Model

**DOI:** 10.3390/insects13070596

**Published:** 2022-06-29

**Authors:** Haopeng Shi, Zhibo Chen, Haiyan Zhang, Juhu Li, Xuanxin Liu, Lili Ren, Youqing Luo

**Affiliations:** 1School of Information Science and Technology, Beijing Forestry University, Beijing 100083, China; tinytree@bjfu.edu.cn (H.S.); zhyzml@bjfu.edu.cn (H.Z.); lijuhu@bjfu.edu.cn (J.L.); liuxuanxin@bjfu.edu.cn (X.L.); 2Engineering Research Center for Forestry-Oriented Intelligent Information Processing, National Forestry and Grassland Administration, Beijing 100083, China; 3Beijing Key Laboratory for Forest Pest Control, Beijing Forestry University, Beijing 100083, China; lily_ren@bjfu.edu.cn (L.R.); yqluo@bjfu.edu.cn (Y.L.)

**Keywords:** trunk-boring beetle, boring vibration, denoising, deep learning, end to end, convolutional recurrent neural network

## Abstract

**Simple Summary:**

Trunk-boring insects belong to one of the most destructive forest pests. Larvae in some groups are particularly difficult to detect since they make their living in trunks and no obvious sign can be found from outside. To deal with this problem, a new method is presented here, which is the embedding of a vibration probe into the tree trunk to pick up vibrations caused by larvae and the use of a model to distinguish whether the tree is infected. However, this procedure can experience severe interference from environmental noise, which is simultaneously picked up with vibrations. Thus, it is necessary to add a noise suppression process before discrimination. Previous examples have proved that the application of the analysis intended for sounds to boring vibrations is feasible. Therefore, we took advantage of deep learning-based speech enhancement and further improved it to develop a boring vibration enhancement model. The training data used in this research contains boring vibrations recorded within pieces of trunks and noise, which are common for trees’ living environment. The experimental results indicate that the enhancement procedure provided by our model substantially increases the accuracy of several well-known classification models, guaranteeing a more practical larvae detection.

**Abstract:**

The larvae of some trunk-boring beetles barely leave traces on the outside of trunks when feeding within, rendering the detection of them rather difficult. One approach to solving this problem involves the use of a probe to pick up boring vibrations inside the trunk and distinguish larvae activity according to the vibrations. Clean boring vibration signals without noise are critical for accurate judgement. Unfortunately, these environments are filled with natural or artificial noise. To address this issue, we constructed a boring vibration enhancement model named VibDenoiser, which makes a significant contribution to this rarely studied domain. This model is built using the technology of deep learning-based speech enhancement. It consists of convolutional encoder and decoder layers with skip connections, and two layers of SRU++ for sequence modeling. The dataset constructed for study is made up of boring vibrations of *Agrilus planipennis* Fairmaire, 1888 (Coleoptera: Buprestidae) and environmental noise. Our VibDenoiser achieves an improvement of 18.57 in SNR, and it runs in real-time on a laptop CPU. The accuracy of the four classification models increased by a large margin using vibration clips enhanced by our model. The results demonstrate the great enhancement performance of our model, and the contribution of our work to better boring vibration detection.

## 1. Introduction

Forests are the principal part of the terrestrial ecosystem and renewable resources of our planet. Forests enhance carbon sequestration, prevent soil erosion and desertification, contribute to the protection of watersheds and air quality, and provide habitats to a diverse array of species [1]. Different forest disasters, including forest fires, tree pathogens, insect pests, and rodents, severely threaten the health of forest ecosystems. They further impact the stable development of agriculture, forestry, and the livelihood of humans [2]. Among forest pest species, trunk-boring beetles are particularly difficult to manage. Such species include *Agrilus planipennis* (Coleoptera: Buprestidae), *Semanotus bifasciatus* (Coleoptera: Cerambycidae), and *Eucryptorrhynchus brandti* (Coleoptera: Curculionidae). They tunnel and feed in the cambium layer of trees, which transports nutrients and water to the leaves. As a result, infested trees become increasingly weaker, their limbs and branches gradually fall off, and they eventually die. What is more troublesome is that some borer infestations often go unnoticed until plants or parts of plants begin to die or show external signs of damage. To deal with this issue, a relatively novel method is to embed a piezoelectric accelerometer into the tree trunk to pick up boring vibrations caused by larvae, and then to feed these vibrations to a trained model to distinguish whether the trunk is infected [3]. To date, the standard equipment for vibration detection using contact sensors is the piezoelectric accelerometer. An accelerometer consists of a piezoelectric crystal coupled with a seismic mass. It detects the displacement of the substrate to which it is attached and measures the acceleration. Regarding its reliability, the highest degree of accuracy is reached by stud mounting the sensor on the substrate [4]. Automatic detection of wood-boring larvae has been investigated before, but the signal analysis has constantly been hampered by background noise that is recorded simultaneously. In order to increase the detectability of vibrations, our research focuses on the preprocessing of this method, which is the enhancement of boring vibration signals.

Several studies have applied the technique described above in the identification of boring vibrations. Bilski et el. [5] used an accelerometer to record the vibrations of wood-boring insects’ larvae in wooden constructions, and then employed the support vector machine to perform classification based on features defined in the time domain. Sutin et al. [6] designed an algorithm that automatically detects the pulses of *Anoplophora glabripennis* and *Agrilus planipennis* larvae with parameters typical for larva-induced signals. The trunk was identified as infected when the mean rate of the detected insect pulses per minute exceeded a predefined threshold. Zhu et al. [7] utilized the sound parameterization technique, which is frequently used in speech recognition, to discern insects. In their study, mel-frequency cepstral coefficients (MFCCs) were extracted from the recordings after preprocessing, followed by classification using a trained Gaussian mixture model (GMM). With the development of deep learning and neural networks, artificial intelligence has been shown to be a promising solution for various challenges that require specialized and labor-intensive work [8]. Many researchers in recent years have held the same point of view and adopted deep learning models in their studies. For example, Sun et al. [9] proposed a lightweight convolutional neural network called InsectFrames, which contains only four convolutional layers. They employed the technique of keyword spotting to automatically identify the boring vibrations of *Semanotus bifasciatus* and *Eucryptorrhynchus brandti* larvae. Karar et al. [10] proposed a new IoT-based framework for early vibration detection of red palm weevils using the fine-tuned transfer learning classifier InceptionResNet-V2, which was trained using vibration data collected by a Tree Vibes [11] recording device.

Although some studies are available, research that has taken the interference of background noise into consideration is rarely seen. However, a particular challenge for the recognition model is the task of detecting relevant signals in the presence of noise. Noise is defined as unwanted sound or signal. There are biotic and abiotic noise sources, such as wind or rain, which is generally below 2 kHz, or anthropogenic noise caused by traffic and heavy machinery [12]. Although the vibrational signaling channel has been traditionally considered “private” and thus is less influenced by environmental noise than the acoustic channel, it can also be highly noisy in plant environments. In the vibrational channel, the frequency range of boring vibrations and the frequency range of noise from the environment overlap, causing severe interference. Studies of the natural vibrational environment show that regardless of the environment studied, geophysical vibrations induced by light wind are nearly always a component of the natural vibroscape that is present. Stronger wind gusts generate high-amplitude vibrations in the frequency range up to 5 kHz, characterized by rapid, unpredictable short-term variations in the amplitude [4]. Our recordings show that the boring vibrations of *Agrilus planipennis* are characterized by frequencies slightly below 2 kHz and about 17 kHz. As depicted in the spectrogram of noise in our recordings, the frequency of birds’ twitter ranges from 1.5 to 4 kHz and the frequency of babble noise ranges from below 400 Hz to above 1 kHz. The amplitudes of boring vibrations are typically low and subjected to masking by incidental noise of a biotic and abiotic origin. In addition to incidental noise, noise from the measurement equipment itself is also contained in the signals. Noise has been verified to have a negative impact on the recognition accuracy [13]. Mankin et al. [14] analyzed the vibrations of *Rhynchophorus* and the background noise in both the time and frequency domain. The results indicated that part of the background noise that has the same frequency as the larval vibrations could interfere with the discrimination of an infestation. Liu et al. [15] designed a recognition model based on the convolutional neural network (CNN) to recognize the boring vibrations of *Semanotus bifasciatus*. Moreover, they tested the noise immunity of the proposed CNN model and GMM. The results clearly showed that noise had a significant impact on the classification accuracy of both the CNN model and GMM: the lower the signal-to-noise ratio (SNR), the greater the decrease in accuracy. When the SNR was −7 dB, the recognition accuracy decreased by 10.8% and 15.6% for the CNN model and GMM, respectively. Zhou et al. [16] introduced improved anti-noise power normalized cepstral coefficients (PNCCs) based on the wavelet package for trunk borer vibrations, and adopted the genetic algorithm support vector machine (GA-SVM) as a classifier. The audio clips in the research consisted of the clean boring vibrations of borer pests of five different species and various kinds of environmental noise. For a −5 dB SNR, the accuracy of the model decreased from 100% to 83%, and a further decline to 70% for −10 dB SNR.

A previous study [5] pointed out that noise that deteriorates proper larva detection should be suppressed if possible. Other research [17] also indicated that environmental noise can be significant and can cover the feeble vibrations of wood-boring insects, consequently leading to false alarms. Therefore, the addition of a denoising or enhancement procedure to boring vibrations can mitigate or even eliminate interference, ensuring accurate early detection and opportune treatment. Yet, most existing discrimination methods of boring vibrations lack a noise suppression procedure or adopt primitive techniques. These techniques include spectral subtraction and the minimum mean square error short-time spectral amplitude estimation in the frequency domain, and adaptive filtering methods in the time domain. Most of the described noise reduction methods require a priori knowledge of the noise profile to operate correctly [13]. Thus, these methods do not yield satisfactory results and are unpractical. With the development of signal analysis in recent years, significant progress has been made. It is now time for them to be applied to biotremology.

To engineers and physicists, both sound and vibration encompass mechanical waves that can be technically described as both vibrational and acoustic. The categorization of these signals in biotremology is biological or perceptual in nature. Sound or acoustic waves are far-field, purely longitudinal waves perceived by pressure or pressure-difference receivers while vibrations are applied in two further ways of emitting mechanical energy in biological interactions, including “contact vibration or rhythmic touch” and “near-field medium motion”. They are perceived by motion detectors. The transmission medium imposes important differences as well. Sounds are air-borne signals and vibrations are substrate-borne signals. Air is a relatively homogeneous medium, and its properties are fairly predictable. On the other hand, substrates are very heterogeneous media, and their transmission properties that influence attenuation and filtering differ depending on the physical properties of the substrate [4]. Despite these differences, boring vibrations may share the same processing technique as acoustic signals. Regarding boring vibration identification models, one of them [9] employed the technique of keyword spotting. Similarly, it is theoretically feasible to apply speech enhancement to boring vibration signals.

Speech enhancement is the task of using noisy speech as input and producing an enhanced speech output for better speech quality, intelligibility, and, sometimes, better criterion in downstream tasks [18]. Since one probe is capable of detecting boring vibrations within a spherical region of a trunk [10], the signal is a single channel, and monaural speech enhancement is the proper technique for this situation. Classical speech enhancement methods include spectral subtraction, Wiener filtering, statistical model-based methods, and subspace methods [19]. Although the above-mentioned algorithms have the capability to suppress background noise, they are cumbersome and complicated [20] and do not generalize well. Recently, neural network-based approaches have experienced much success in speech enhancement due to their powerful modeling capabilities [21]. Among the neural network-based methods, a portion of them carry out enhancement on frequency-domain acoustic features, which are called spectral-mapping-based approaches. In these approaches, speech signals are analyzed and reconstructed using the short-time Fourier transform (STFT) and inverse STFT, respectively. Another class of methods directly perform enhancement on the raw waveform, which are called waveform-mapping-based approaches [22]. The waveform-mapping-based approaches do not rely on the representation of speech signals in the frequency domain, and as a result avoid the loss of accurate phase information. In addition, it is a simpler procedure due to the cancellation of unwanted signal transformation between the time domain and frequency domain.

This study takes the boring vibrations of emerald ash borer (EAB) larvae as the research subject. The emerald ash borer, *Agrilus planipennis* Fairmaire, 1888 (Coleoptera: Buprestidae), is an invasive beetle of East Asian origin that has caused extreme levels of mortality in ash [23], with a devastating economic and ecological impact [24]. The monitoring methods for EAB are mainly visual inspections and the application of pheromone and color traps. The control measures include the cutting of infested trees, which are mostly detected by dry branches and typical D-shaped exit holes on the bark. In addition, the replacement of North American and European ash trees with more resistant Asian ash species or possibly hybrids, and chemical and biological control is available [25]. The detection of trunk-boring beetles by their vibrational cues is an efficient and convenient approach. It is independent of visual access. Thus, it is capable of an early warning and early reaction. The aforementioned necessity for a denoising or enhancement process for the boring vibration signals of Buprestidae encouraged us to propose a waveform-mapping-based boring vibration enhancement model called VibDenoiser (Vibration Denoiser), which consists of convolution layers and SRU++ [26] layers. The dataset used in this research consists of boring vibrations and environmental noise and a mixture of them. For boring vibrations, we inserted a piezoelectric vibration probe into several ash trunks that were infected by EAB larvae to pick up their boring vibrations. For environmental noise, we inserted the same probe into a dead ash trunk that was not infected by EAB and placed the trunk in noisy environments to pick up noise propagated in the trunk. Our results showed that VibDenoiser is able to increase SNR by 18.57, and it runs in real-time on a single laptop CPU core. We applied our noisy boring vibrations to four well-known classification models, namely VGG16 [27], ResNet18 [28], SqueezeNet [29], and MobileNetV2 [30]. Their classification accuracies were 81.14%, 89.39%, 78.45%, and 85.77%, respectively. It is gratifying that their accuracies increased by a large margin, reaching 92.51%, 96.47%, 88.89%, and 90.40% using vibration clips enhanced by our model. These results prove that VibDenoiser is able to suppress noise effectually with an affordable expense, ensuring a more accurate early detection of larvae. 

## 2. Materials and Methods

### 2.1. Dataset Preparation

#### 2.1.1. Data Collection and Filtering

All boring vibrations and environmental noise used in this research were collected in the course of this investigation. We selected and cut down several ash trees that were in different conditions, including alive, dying, and dead. The trees are from an EAB larvae-infected forest farm located in Tongzhou District, Beijing. All the tree branches were removed, and the trunks cut into sections of an equivalent length. After the selection, we acquired 6 trunk sections from live trees, 4 from dying trees, and 2 from dead trees. A piezoelectric vibration probe was inserted into the trunk sections about 2 cm deep to pick up locomotion and boring vibrations. The probe we utilized was jointly developed by Beijing Forestry University and Beihang University. To calibrate the probe, we used a signal generator to generate a sinusoidal wave with a frequency between 500 and 10k Hz to simulate boring vibrations. The generated signals were then transmitted to a power amplifier and amplified for a vibration exciter. The probe was mounted on the vibration exciter to pick up vibrations. Finally, an oscilloscope displayed the original signal and the signal detected by the probe for comparison. The sampling rate for recording was 44,100 Hz and the bit depth was 16 bit. The signals were transmitted to a computer and were saved in wave format using Audacity. Since boring vibrations are feeble, the recording procedure was conducted in an unoccupied laboratory with no other activities. The computer, probe, and trunk were placed on a normal table. Before recording, we monitored the larvae activity using headphones to confirm it was active. Once the recording was started, all individuals left the room. For trunks from living trees, we recorded the vibrations inside for one and a half hour each day if the larvae in it were still active. Specifically, multiple audio clips for one trunk section were recorded at different times. For trunks from dying and dead trees, they barely showed any larvae activity, and most of the recording contained only noise floor in the instrumentation. Therefore, one audio clip for each trunk section was adequate. The recording started on 23 July 2021 and ended on 27 July 2021. About 6 segments were recorded each day, with 5 segments for each trunk. To ensure all vibrations were caused by EAB larvae, we peeled the bark of all trunk sections after the whole recording procedure to reveal and count the EAB larvae inside the trunks under the supervision of forestry professionals.

After collecting the audio clips, we trimmed the start and end of the clips since these parts mainly contained noise caused by manual operations. Then, we inspected their spectrogram for any abnormal noise floor and unwanted noise such as footsteps, rain drops, distant speech, and sudden burst caused by instruments, and removed it manually from the spectrogram. As for the audio clips from the live trees, we further discarded whole clips that indicated inactive larval activity.

Environmental noise was also needed to produce a mixture with boring vibrations in this study. We chose five locations to record noise where it is common for the environment to contain trees and reasonable noise. Four of them are located in Beijing Forestry University and one in Olympic Forest Park. The geographic coordinates are 40°1′4″ N, 116°23′47″ E for the site in Olympic Forest Park and 40°0′26″ N, 116°21′2″ E, 40°0′28″ N, 116°21′13″ E, 40°0′27″ N, 116°21′14″ E, and 40°0′32″ N, 116°21′15″ E for the sites in Beijing Forestry University. To ensure consistency with the boring vibrations, we inserted the same probe into a dead ash trunk section that was not infected by EAB and used the same computer to record the noise propagating in the trunk. The majority of the noise clip contained wind noise, the rustling of leaves, birds’ twitter, babble noise, footsteps, and tire noise. In order to improve the training efficiency, we removed all segments that did not include any noise but noise floor.

#### 2.1.2. Dataset Production

To eliminate the influence of the noise floor, we implemented spectral subtraction [31] in Python and applied it to the recordings. Spectral subtraction requires a segment of the sample as the target. In this case, the sample is a segment of the noise floor. For the boring vibration recordings, a segment of the recording from dead trunk was chosen as the noise floor sample. The noise floor samples used for the environmental noise are relatively quiet segments from the environmental noise. For the sake of easier processing afterwards, we split all recordings into 5 s segments. In the experiment, 94 percent of the data was used for training, and the other 6 percent was used for testing purposes. Thus, 94 percent of the boring vibration segments and environmental noise segments were selected to construct the training set, and the remaining were used to construct the test set. We randomly mixed the boring vibration segments and environmental noise segments at an SNR of −10 for both the training set and test set. Due to hardware limitations, we selected half of the audio clips that contained the most energy [32] as our dataset. As a result, the training set contained 9940 clips, with about 13.8 h of boring vibrations, and the test set contained 632 clips, with about 52.67 min. To minimize the likelihood of model overfitting and selection bias, the audio in the test set was not seen by the model during training.

### 2.2. VibDenoiser Architecture

#### 2.2.1. Acoustic Signal Enhancement Model

A wide variety of deep learning-based speech enhancement models have emerged in recent years and contributed to significant accomplishments in the field. Among these methods, one that balances the enhancement effect and inference speed is the waveform-mapping-based model with an encoder-decoder architecture and convolutional recurrent neural network (CRN) [33]. Compared to a fully connected architecture, fully convolution layers retain local information better, and thus can more accurately model the frequency characteristics of the waveform [22]. With the knowledge above, a newly proposed method, also known as DEMUCS [34], was constructed. It consists of an encoder and decoder made up of multiple convolution layers with skip connections across them similar to those in U-Net [35]. Between the encoder and decoder sits two layers of the sequence modeling module LSTM, which take the encoders’ output and passes the information to the decoder after its processing. DEMUCS is made for monaural speech enhancement and can operate in real-time applications. Due to the specification above, we found the DEMUCS suitable for our purpose and further fit it into our end-to-end VibDenoiser.

Specifically, the encoder of VibDenoiser takes a boring vibration segment in the raw waveform as input and outputs the encoded latent representation of the segment. Altogether, there are L encoder layers. A single encoder layer consists of two convolution layers and two activation layers following each convolution layer. The first convolution layer has a kernel size of K and stride of S and $2^{i-1}\times C_{h}$ output channels, where i represents the layer number of the present encoder layer. Then, an ReLU activation [36] is straight after the first convolution layer. The second convolution layer is a “1×1” convolution whose input channel is the same as the output channel of the prior layer and doubles its output channel to $2^{i}\;\times \;C_{h}$. Followed by a GLU [37] activation, the number of channels is decreased to $2^{i-1}\;\times \;C_{h}$ as the final output of an encoder layer.

As for the sequence modeling module, we stacked two layers of SRU++ [26], an improved version of SRU [38]. SRU yields a performance that is comparable to LSTM, but the time complexity is much lower [38]. Here, SRU++ takes the latent representation of the segment and performs the enhancement in the latent space. The output is the same size as the input in this layer.

Finally, the decoder takes the output of the sequence modeling module and decodes the enhanced latent representation back to waveform data. The decoder is symmetric to the encoder layer in general. A single decoder layer also contains two convolution layers, each followed by one activation layer. The first layer is a “1×1” convolution layer, which takes a $2^{L-i}\;\times \;C_{h}$ channel input and output $2^{L-i+1}\;\times \;C_{h}$ channels. The second layer is a GLU activation layer, and the channel of data is decreased by 2 after this layer. The third layer is the transposed convolution with a kernel size of K, stride of S, and $2^{L-i}\;\times \;C_{h}$ input channels and $2^{L-i-1}\;\times \;C_{h}$ output channels followed by an ReLU activation layer as the fourth layer. Note that the last decoder layer has a single channel output and there is no ReLU activation applied. Moreover, skip connections are employed to transfer the output of the i-th encoder layer to the input of the L−i+1-th decoder layer. The detailed structure of the encoder and decoder layers is shown in Figure 1, and the overall structure of VibDenoiser is shown in Figure 2.

#### 2.2.2. Simple Recurrent Unit

The simple recurrent unit (SRU) [38] is a simplified recurrent architecture that offers both high parallelization and sequence modeling capacity. It has an ideal trade-off between the scalability and representational power as well. It was first introduced and applied in natural language processing. Nevertheless, it has been successfully fitted into speech enhancement models, such as [20,22], suggesting its further application in the field.

A single layer of SRU consists of the following computation:(1)$$f_{t}=\sigma \left(W_{f}x_{t}+v_{f}⊙c_{t-1}+b_{f}\right),$$
(2)$$c_{t}=f_{t}⊙c_{t-1}+\left(1-f_{t}\right)⊙\left(Wx_{t}\right),$$
(3)$$r_{t}=\sigma \left(W_{r}x_{t}+v_{r}⊙c_{t-1}+b_{r}\right),$$
(4)$$h_{t}=r_{t}⊙c_{t}+\left(1-r_{t}\right)⊙x_{t}.$$

Among them, $f_{t}$, $c_{t}$, $r_{t}$, $h_{t}$, $c_{t-1}$, and $x_{t}$ stand for the forget gate, internal state, reset gate, output state, previous state, and current input, respectively. W, $W_{f}$, and $W_{r}$ are parameter matrices; $v_{f}$, $v_{r}$, $b_{f}$, and $b_{r}$ are parameter vectors that are learnable during training; $b_{f}$ is the bias term of the forget gate and $b_{r}$ is the bias term of the reset gate; σ(·) is the sigmoid function; and ⊙ represents point-wise multiplication. Equations (1) and (2) comprise the light recurrence part, and Equations (3) and (4) comprise the highway network part. Figure 3 demonstrates the structure of SRU.

The light recurrence part takes the current input and previous state to compute the forget gate, which controls the information flow (Equation (1)). Then, the forget gate modulates the internal state by adaptively blending the previous state and the current input (Equation (2)). The highway network part boosts gradient-based training of deep neural networks. In order to combine the current input and the internal state, the reset gate is added to the computation, and it shares a similar calculation with the forget gate (Equation (3)). The calculation of the final output utilizes the reset gate to adaptively combine the current input and the current state. Under the adaptive combination of the current state, which is produced by the light recurrence part and the current input using the reset gate, the final output is computed as Equation (4). One novel design is the change from matrix multiplication to point-wise multiplication between the parameter and $c_{t-1}$. The use of matrix multiplication renders it difficult to parallelize due to the need for all entries of $c_{t-1}$ for the computation. In other words, the computation of each dimension of $c_{t}$ and $f_{t}$ has to wait for a complete $c_{t-1}$. With the help of point-wise multiplication, each dimension of $c_{t-1}$ becomes independent, thus enabling more parallelization. Furthermore, the second term in Equation (4) is a skip connection, which allows the gradient to directly propagate to the previous layer. This particular design has been verified to improve model scalability [28,39].

The parallelized implementation also contributes to the fair speed of SRU. Two optimizations are performed in the context of CUDA programming to enhance parallelism. The first one is the calculation of the matrix multiplications in Equations (1)–(3) across all time steps simultaneously, as shown in Equation (5). L stands for the sequence length and U is the computed matrix. This optimization significantly improves the utilization of GPU:(5)$$U^{T}=\left(\begin{array}{c} W\\ W_{f}\\ W_{r} \end{array}\right)\left[x_{1},x_{2},\ldots ,x_{L}\right].$$

The second one is that all point-wise operations are compiled into a single fused CUDA kernel and the computation of each dimension of the hidden state is parallelized.

On the basis of SRU, SRU++ [26] further incorporates the attention mechanism for better modeling capacity. Specifically, the linear transformation of the input X to obtain U in Equation (5) is replaced by the self-attention operation. The computation of the query, key, and value matrices of the self-attention component is shown in the following equations:(6)$$Q=W^{q}X^{T},$$
(7)$$K=W^{k}Q,$$
(8)$$V=W^{v}Q.$$
where $W^{q}$, $W^{k}$, and $W^{v}$ are parameter matrices; and X is the input sequence. The dimension of $W_{k}$ and $W_{v}$ is much smaller than the input. The weighted average output A is computed using the scaled dot product attention [40] afterwards:(9)$$A^{T}=\mathrm{softmax}\left(\frac{Q^{T}K}{\sqrt{d^{\prime }}}\right)V^{T}$$
where $d^{\prime }$ is the attention dimension, in other words, the dimension of the $d^{\prime }\times d^{\prime }$ matrices $W_{k}$ and $W_{v}$. U is obtained by a final linear projection with layer normalization after the attention operation:(10)$$U^{T}=W^{o}\mathrm{layernorm}\left(Q+\alpha \cdot A\right).$$

In Equation (10), α is a learnable scalar, which is initialized as zero; and $W^{o}$ is a parameter matrix. Q+α⋅A contains a residual connection, which is beneficial to gradient propagation and helps stabilize the training process.

#### 2.2.3. Loss Function

We tested six different loss functions, namely L1 loss, L2 loss, Huber loss, L1+SNR loss, L1+STFT [34] loss, and log-cosh loss. The loss functions include calculation in both the time domain and frequency domain, as listed below:(11)$$L_{l1}=\sum \left|y-\hat{y}\right|,$$
(12)$$L_{l2}={\sum \left(y-\hat{y}\right)}^{2},$$
(13)$$L_{huber}=\{\begin{array}{ll} \frac{1}{2}{\left(y-\hat{y}\right)}^{2}, & if\;\left|y-\hat{y}\right|<1\\ \left|y-\hat{y}\right|-\frac{1}{2}, & otherwise \end{array},$$
(14)$$L_{l1+SNR}{=\mathrm{logL}}_{l1}-SNR\left(y,\hat{y}\right),$$
(15)$$L_{l1+stft}=L_{l1}+\sum_{i=1}^{M}L_{stft}\left(y,\hat{y}\right),$$
(16)$$L_{\log\cosh}=\sum \log\left(\cosh\left(\hat{y}-y\right)\right).$$
where the symbols y and $\hat{y}$ are the clean signal and the enhanced signal, respectively. The STFT loss is introduced in [34]. There are examples that use log-cosh as the loss function for the speech enhancement model, one built with GRU [41] and the other built with SRU [20]. We creatively used one of the evaluation metrics, specifically SNR, as a term in the loss function. However, our experimental result showed that log-cosh is the best performing loss function. We adopted log-cosh as the loss function for VibDenoiser. The detailed results of our experiment on different loss functions are given in the results section.

## 3. Results

### 3.1. Experimental Settings

VibDenoiser was built on the basis of Python and PyTorch [42] with the network settings of L = 5, K = 8, S = 4, and $C_{h}$ = 48. All model parameters were initialized using the method proposed in [43]. We trained our model with the Adam optimizer [44] and a learning rate of 3 × 10^−4^. The total training epoch was 150 epochs, and we evaluated every 5 epochs. Different batch sizes were set for the model with different SRU++ layer numbers due to hardware limitations. The boring vibrations were recorded with a sample rate of 44,100 Hz and a bit depth of 16 bit, monaural. All inputs were normalized by the standard deviation before feeding them into the model. The hardware platform of our experiment included a workstation with Intel Xeon Gold 5120 and four NVIDIA Tesla T4.

### 3.2. Evaluation Metrics

We evaluated our model with three commonly used objective metrics, including the signal-to-noise ratio (SNR), segmental signal-to-noise ratio (SegSNR), and log-likelihood ratio measure (LLR) [45]. We did not employ the commonly used Perceptual Evaluation of Speech Quality (PESQ) and Short-Time Objective Intelligibility (STOI) because they include perceptual quality assessment of acoustic signals for humans, which is unreasonable since the boring vibration is enhanced for discrimination models. The computational methods are shown in the following formulas:(17)$$SNR{=10\log}_{10}\frac{\sum_{n=0}^{N-1}s^{2}\left(n\right)}{\sum_{n=0}^{N-1}\left(s^{2}\left(n\right)-{\hat{s}}^{2}\left(n\right)\right)},$$
(18)$$SegSNR=\frac{10}{M}\sum_{m=0}^{M-1}\log_{10}\frac{\sum_{n=Nm}^{Nm+N-1}s^{2}\left(n\right)}{\sum_{n=Nm}^{Nm+N-1}\left(s^{2}\left(n\right)-{\hat{s}}^{2}\left(n\right)\right)}.$$
where s represents clean signal and $\hat{s}$ represents enhanced signal. M is the number of frames in a segment of signal, N is the number of samples in Equation (17), and the frame length in Equation (18). The LLR measure is defined as:(19)$$LLR\left(a_{x},{\bar{a}}_{\hat{x}}\right)=\log\frac{{\bar{a}}_{\hat{x}}^{T}R_{x}{\bar{a}}_{\hat{x}}}{a_{x}{}^{T}R_{x}a_{x}},$$
where $a_{x}{}^{T}$ is the LPC coefficients of the clean signal, ${\bar{a}}_{\hat{x}}^{T}$ is the coefficients of the enhanced signal, and $R_{x}$ is the autocorrelation matrix of the clean signal.

### 3.3. Boring Vibration Enhancement Results

First, we tested different layers of SRU++ in our model and left the rest of model structure the same as depicted in Section 2.2.1. We started from a 14-layer SRU++, as it had a similar model size to DEMUCS with 2-layer LSTM. Then, we tried a 4-layer SRU++. Unsurprisingly, it performed as well as the 14-layer SRU++ but with a much smaller model size. Theoretically, as long as the layers of SRU++ are not excessive, the more the better. Hence, we added two layers at a time and experimented several times. The 8-layer SRU++ showed the best performance. Table 1 presents the results in terms of SNR, SegSNR, and LLR. More details of the performance of our model with different SRU++ layers are shown in Table A1 in Appendix A.

We trained the VibDenoiser with 8-layer SRU++ for 150 epochs. The results of the evaluation during training demonstrated that the best state was achieved at about epoch 50. After this, the model seemed to have over-fitted according to the drops in the metrics. Thus, all results used for comparison were from the evaluation of the model trained after 50 epochs.

Then, we trained our model with 8-layer SRU++ using 6 different loss functions and compared the results provided in Table 2. It clearly illustrated that the time domain point-to-point losses, including L1, L2, Huber loss, and log-cosh loss, performed much better than frequency domain loss. The SNR loss proposed by us did not reach our expectation. Surprisingly, the best loss function in our experiment was the log-cosh.

After determining the optimal loss function, we further profiled our model with different SRU++ layers. A detailed analysis of the inference time and model size is shown in Table A2 and Table A3 in Appendix A. The 8-layer SRU++ model had a slightly longer inference time compared to the 2-layer LSTM model when using CPU, which is unacceptable considering the purpose of VibDenoiser. In order to alleviate this problem, we conducted additional experiments on the model with 2-layer SRU++. Table 3 presents the results of the 2-layer and 4-layer SRU++ model using different loss functions. Surprisingly, the 2-layer SRU++ model showed a shorter inference time and a smaller model size at the expense of a slight decrease in the enhancement performance compared to the 8-layer SRU++ model. As illustrated in Table 4, the required GPU hours for training decreased as well. The training times of different models are shown in Table A4 in Appendix A. Figure 4 shows several enhancement results of VibDenoiser. The noise contained in the segments of column a is birds’ twitter, wind noise, babble noise, and the rustling of leaves, from top to bottom. Most of the EAB vibrations were masked by noise and were undistinguishable. It is clearly shown in the spectrograms that the background noise was suppressed, leaving only the vibration of EAB.

### 3.4. Classification Results

We applied four well-known classification models to noisy boring vibrations and boring vibrations enhanced by VibDenoiser, respectively, to check the increase in the classification accuracy. The models tested for the classification included VGG16, ResNet18, SqueezeNet, and MobileNetV2. The training data used for the classification models consisted of two classes, which were infected and uninfected. The infected class contained 891 boring vibration segments from noisy training and 891 segments from clean training of our dataset for VibDenoiser. Regarding the uninfected class, 840 segments of noise for the generation of noisy training and 940 segments of recordings from a dead trunk were used. The number of boring vibration segments in both classes was about the same so that the two classes were balanced. We trained the four selected classification models with the smaller dataset described above. Then, we tested their classification accuracy on both the test set and the enhanced test set. The enhanced test set was generated by employing VibDenoiser to enhance the segments of the infected class in the test set. Table 5 provides the classification results. The accuracy increased substantially on all four models using the enhanced test set. This is direct proof of the excellent denoising ability of VibDenoiser.

## 4. Discussion

Until now, convenient and efficient monitoring of trunk-boring beetles has remained a difficult problem in pest control and forest management. An appropriate solution is to record vibrations in tree trunks by embedding a piezoelectric accelerometer and use a trained model to detect boring vibrations in these recordings. As stated in a previous study, environmental noise can be significant and cover the feeble vibrations of wood-boring insects [17], thus having a negative impact on the recognition accuracy [13]. It is necessary to add an enhancement procedure to the boring vibrations before recognition. Inspired by a previous study [9], the method for acoustic signals is applicable to biological signal analysis. Specifically, the use of methods for processing air-borne sound signals is also feasible for substrate-borne boring signals. Considering the necessity for the enhancement procedure, deep learning-based speech enhancement is the best alternative to realize it. Deep learning-based speech enhancement has shown numerous breakthroughs in recent years. Its powerful modeling capability may enhance signals beyond speech. An approach to detecting larvae activity relies heavily on clear boring vibration signals. Aiming to alleviate this problem, our research successfully applied speech enhancement to the boring vibrations of trunk-boring larvae. Despite this interest, as far as we know, almost no researchers have applied enhancement models to boring vibrations. We recorded dozens of hours of EAB boring vibrations and environmental noise to create our dataset. The enhancement model VibDenoiser was proposed by us, employing deep learning-based speech enhancement. VibDenoiser is a waveform-mapping-based model with an encoder-decoder architecture and convolutional recurrent neural network (CRN). The encoder and decoder both consist of five convolution layers, and two layers of SRU++ are placed between them, functioning as sequence modeling module. The loss function for our model is log-cosh, which is a time domain point-to-point loss. It resembles the mean square error but is not susceptible to abnormal points. We also tested the frequency domain loss function, but it yielded a poor result. This was probably caused by the time domain evaluation metrics we adopted. The model with 8-layer SRU++ showed the best performance in our study. However, the model size and inference time were not satisfying and can be further reduced. The 2-layer SRU++ substantially reduced the model size and inference time at the expanse of a slight decrease in the enhancement effect, which is more in line with the practical application. It was clearly seen in the frequency spectrum of both the noisy and enhanced boring vibration segments (Figure 4) that VibDenoiser could suppress most of the noise, leaving clear vibrations of EAB for classification. We applied four well-known classification models to both noisy and enhanced boring vibration segments for discrimination, and the accuracy was increased by a large margin. This further proves the necessity of an enhancement procedure and the excellent performance of VibDenoiser.

Datasets in future research studies should include boring vibrations of a variety of larvae to train models with improved universal applicability. Future model development should aim for lighter models with less parameters and faster inference. Our model is rather valuable for the development of insect pest monitoring. It can be integrated into larvae surveillance programs with mobile deployment for general use. Prototypes of trunk-boring beetle monitoring system built with our model are capable of early warning in both forests and urban areas; therefore, this enables early reaction and treatment, such as quarantine measures, sanitation felling, chemical control, and biological control [25]. Some parasitoids are potential agents for the biological control of EAB. For example, a braconnid *Spathius polonicus* Niezabitowski (Hymenoptera: Braconidae: Doryctinae); a species of egg parasitoid *Oobius* sp. (Hymenoptera: Encyrtidae); and three species of larval ectoparasitoids: *Tetrastichus planipennisi* Yang (Hymenoptera: Eulophidae), *Atanycolus nigrivensis* Voinovskaja-Krieger (Hymenoptera: Braconidae: Braconinae), and *Spathius galinae* Belokobylskij et Strazanac (Hymenoptera: Braconidae: Doryctinae) [46]. Chemical (insecticide) control of EAB falls into three categories: systemic insecticides that are applied as soil injections or drenches; systemic insecticides applied as trunk injections or trunk implants; and protective cover sprays that are applied to the trunk, main branches, and foliage. Insecticide formulations include Merit^®^, IMA-jet^®^, Imicide, and some other options [47]. We hope that our research study is conducive and facilitates the development of a non-invasive early detection tool for larvae infestations. Consequently, it contributes to pest control and forest management.

## Figures and Tables

**Figure 1 insects-13-00596-f001:**
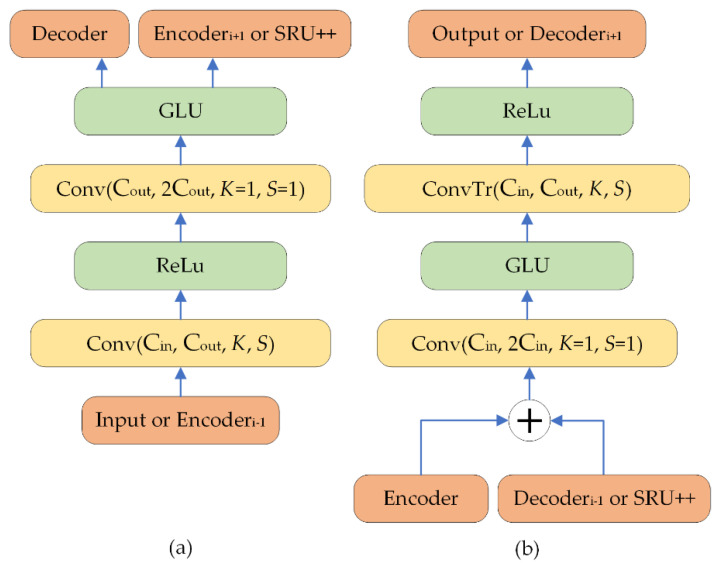
Detailed architecture of the encoder (**a**) and decoder (**b**) of VibDenoiser, where K stands for the kernel size and S stands for the stride. The arrows represent the flow of information.

**Figure 2 insects-13-00596-f002:**
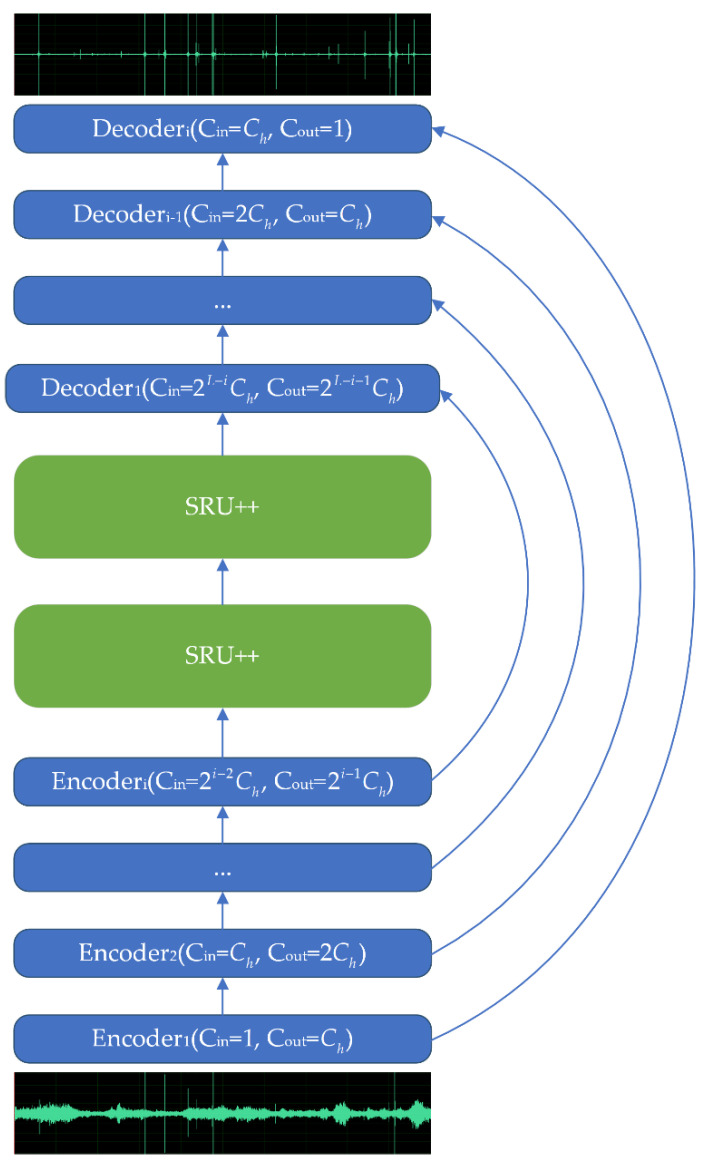
The architecture of the proposed VibDenoiser model. The arrows represent the skip connections. $C_{h}$ is the number of channels and L is the layer number.

**Figure 3 insects-13-00596-f003:**
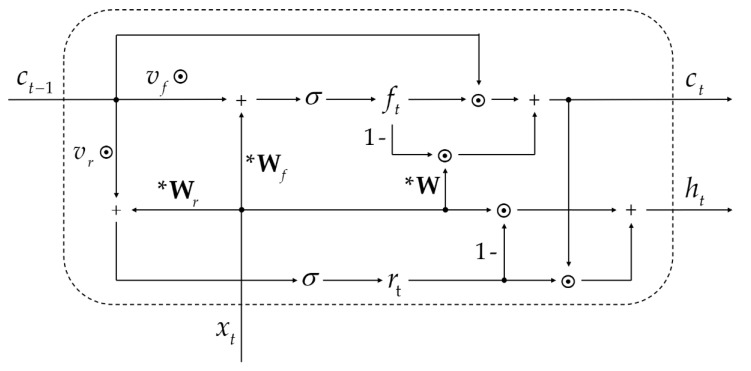
The architecture of SRU. $x_{t}$ is the input vector, $C_{t}$ is the internal state, $c_{t-1}$ is the previous state, $f_{t}$ is the forget gate, $r_{t}$ is the reset gate, and $h_{t}$ is the output state. * represents matrix multiplication.

**Figure 4 insects-13-00596-f004:**
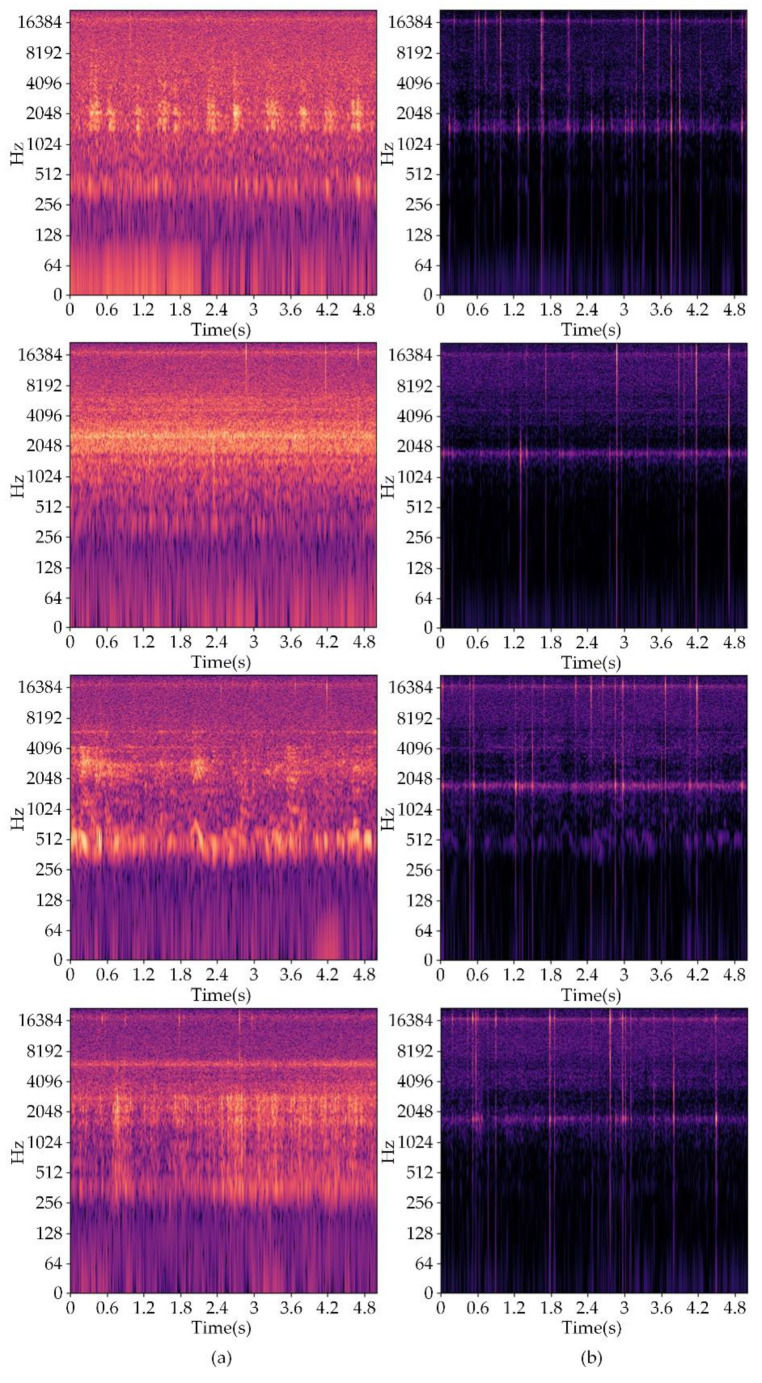
Frequency spectrums of 4 noisy boring vibration segments (column **a**) and the frequency spectrums of the same segments after enhancement using VibDenoiser (column **b**).

**Table 1 insects-13-00596-t001:** Comparison of the model’s performance with 2-layer LSTM, 4-layer, 8-layer, and 14-layer SRU++ at 150 epochs.

Recurrent Network	SNR (dB)	SegSNR (dB)	LLR
2-layer LSTM	8.23	0.50	0.38
4-layer SRU++	8.43	0.62	0.43
8-layer SRU++	8.52	0.65	0.45
14-layer SRU++	8.43	0.63	0.37

**Table 2 insects-13-00596-t002:** Model’s performance with 8-layer SRU++ under different loss functions. The training epoch was 50 and the batch size was 56.

Loss Function	SNR (dB)	SegSNR (dB)	LLR
L1	8.62	0.70	0.40
L2	8.59	0.65	0.36
Huber	8.54	0.67	0.36
L1 + STFT	6.61	−1.38	0.11
L1 + SNR	7.56	0.43	10.48
log-cosh	8.65	0.71	0.40

**Table 3 insects-13-00596-t003:** Comparison of the model performance with 2-layer, 4-layer SRU++, and different loss functions. The training batch size was 56 and the epoch was 50.

Recurrent Network	Loss Function	SNR (dB)	SegSNR (dB)	LLR
2-layer SRU++	L1	8.49	0.65	0.31
2-layer SRU++	log-cosh	8.57	0.66	0.33
4-layer SRU++	log-cosh	8.55	0.68	0.37

**Table 4 insects-13-00596-t004:** The training GPU hours for the model using 2-layer LSTM, 2-layer SRU++, and 8-layer SRU++ at 150 epochs on 4 GPUs.

Recurrent Network	GPU h
2-layer LSTM	90.48
2-layer SRUPP	77.04
8-layer SRUPP	86.48

The batch size for the 2-layer LSTM, 2-layer SRU++, and 8-layer SRU++ model was 64, 56, and 56, respectively.

**Table 5 insects-13-00596-t005:** Classification results of four well-known classification models on the test set and VibDenoiser enhanced test set, respectively.

Classification Model	Accuracy on Noisy Test Set	Accuracy on Enhanced Test Set
VGG16	81.14%	92.51%
ResNet18	89.39%	96.47%
SqueezeNet	78.45%	88.89%
MobileNetV2	85.77%	90.40%

## Data Availability

The data presented in this study are available on request from the corresponding author.

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
