# Peer review of "A Waveform Mapping-Based Approach for Enhancement of Trunk Borers’ Vibration Signals Using Deep Learning Model"

_insects, 2022, doi:10.3390/insects13070596_

Round 1

Reviewer 1 Report

In the manuscript „A Waveform Mapping-Based Approach for Enhancement of Trunk Borers’ Vibration Signals Using Deep Learning Model“, H. Shi et al. present an analysis system detecting beetle vibrational cues generated in tree trunks, and therefore outside of visual access. The description of the set-up processing based on deep learning is rather technical but allows to address important issues for pest control, forest management, and biological signal analysis. Reliable detection of pest insects by vibrations would provide an important diagnostic tool. Based on original recordings, different loss functions are tested for resulting signal recognition, and allow to evaluate for the highest gain of signal-to-noise ratio. The main finding is the reliable vibration detection with a high SNR. The presentation is concise and aided by a clear structure and figures which demonstrate the results from the analysis process, though the manuscript would benefit from corrections by a native speaker.

Major:

In general for publications, the use of terminology between “sound” and “vibration” can be confusing to readers, and should be carefully distinguished (although the terminology also allows for some continuity). The authors state that they recorded “boring vibrations” (L 131) but also “feeding sounds” (L 153). Are these different signals/cues? The introduction also notes the recording of “sound” (L 60, 66) in other publications. It would be helpful to distinguish in the text between airborne sound when recorded with a microphone, and substrate vibrations or wood vibrations when recorded from the solid by an accelerometer or piezo probe. Also, consider mentioning the role of piezo elements in the study of biological signals and their reliability, as by Nieri et al. (2022) Inexpensive Methods for Detecting and Reproducing Substrate-Borne Vibrations: Advantages and Limitations, in Hill et al. (eds) Biotremology: Physiology, Ecology, and Evolution, Springer, pp. 203-218.

The problem of noise for substrate vibrations is only mentioned rather generally (L 81ff, L 174ff) but can hardly be underestimated for effects on animal communication, ecosystem signatures, and technical analysis of signals/cues. Consider including some reference on the role of noise affecting the vibrational channel, e.g. Cocroft and Rodriguez (2005) BioScience 55: 323–334, or Korinšek et al. (2019) Automated Vibrational Signal Recognition and Playback, in Hill et al. (eds) Biotremology. Studying Vibrational Behavior, Springer, pp. 149-173. The complex natural vibroscape including background noise, and also different types of recording tools, are reviewed in Strauß et al. (2021) Advances in Insect Physiology 61: 189-307.

In the current form, the discussion is rather short, and highlights the different applications of the analysis system. In the long reach, maybe mention or at least reference what follows from the identification of EAB in a forest, how could the pest be contained once confirmed? Does the system allow for specific, early reactions? For a broad journal like “Insects”, these aspects would be particularly interesting. Can this system eventually be used to monitor trees before cutting them down, as a “non-invasive” analysis tool? It would be helpful to find these aspect spelled out in the discussion.

The reference section cites a high proportion of conference proceedings papers, are further full research publications available?  

Fig. 4 should include axes and dimensions for the spectrum segments.

Minor:

L 26     to focus the interest of readers on the main problem, rearrange the sentence: “the detection of larvae of trunk-boring beetle is technically difficult, since some barely leave traces…”

L 35     species’ taxonomic information: include also year of description

L 39     last sentence of abstract can be excluded.

L 48     This sentence seems redundant: “Forest disasters… are devastating to forests.” If you want to maintain this, rewrite e.g., “Different forest disasters including [examples] severely threaten the health of forest ecosystems” or similar.

L 50     include the species taxonomic name and the insect group here.

L 57     This description reads as if the species in this paper is identical to the one in the present study. Only the reference title indicates it is another one, as also the species studied here is only introduced in detail on the following page: Please clarify in the text. 

L 96     A previous study…

L 138f  This section already contains results.

L 148   “…collected in the course of this investigation.”

L 153   as the probe was apparently not mounted to the outside of the tree, how deep was it inserted?

L 205   “…came into our sight”? – or simply: “…was constructed.”

L 207   “…sit two layers…

L 213   waveform?

L 305   The two symbols appear identical in the manuscript file.

L 317   “The total training was 150 epochs…”

L 319   44100 Hz

L 371   can you identify the different sources of noise in the recordings? Would they have masked the EAB vibrations to leave them undistinguishable in the recordings?

L 454   lacks the source title

L 464   publication title “insect seconds”: should be “insect sounds”

L 491   species name should be set in italics

Author Response

I have referenced and understood the differences of acoustic signals and boring vibrations. All "sound" in manuscript are changed to "vibration".

The role of  piezoelectric accelerometer are added at line 66 to 71. The interference of noise to the vibration channel is depicted in the third paragraph of Introduction, and more are addded according to the recommended references.

I add the monitoring methods and control measures of EAB to both the Introduction and Discussion. The efficiency and convenience of the relatively new recognition method is more clearly shown, compared to existing method. The new recognition system is built to be capable of giving early warning. Early reactions is therefore more practical. Our model is very useful for building such a recognition system. The answers about applications of the new system are added to the Discussion section.

During the revision, several references of full research publications and book  are added.

Figure 4 was redrawn with axes and dimensions.

All minor issues are fixed except "L 138f  This section already contains results." I failed to see the problem with it.

Reviewer 2 Report

The manuscript “A Waveform Mapping-Based Approach for Enhancement of Trunk Borers’ Vibration Signals Using Deep Learning Model” is an interesting study that aims to apply the deep learning-based speech enhancement methodology to increase the detectability of vibrations produced by wood-boring beetles in audio files recorded in natural or noisy environment. To build and test the methodology, the authors used audio files recorded with a piezo probe inserted into ash trees infested by Agrilus planipennis larvae.

Applied biotremology, the use of substrate-borne vibrations for pest management, is a relatively novel field of study that is lately getting increased interest in the scientific community. The possibility to automatically detect vibrations induced by chewing insects in wood or stored grains has been foreseen for a long time, but the technology, and the techniques of signal analysis in particular, are now on the rise and time has come for them to be applied also to biotremology. In this context, the proposed study is extremely interesting and deserve to be published. However, I am not convinced that the manuscript, as it is, fits the journal Insects. Almost the entirety of the manuscript deals with the software architecture and the algorithm structure and mechanism of action. The authors developed and tested a model for data analysis and the terminology used throughout the text is closer to modelling than entomology. For this reason, even if the model developed, VibDenoiser, can be extremely useful in the development of prototypes for insect pest monitoring, my final suggestion is to reject the article for publication in Insects, but I strongly encourage the authors to submit the manuscript to a more technical journal that would better fit the study (such as Acoustics, Algorithms, Modelling, Analytics).  

Besides my comment on the appropriateness of the manuscript for publication in Insects, to improve the quality of the manuscript, I would encourage the authors to clarify some concepts and sections. For instance, more than once vibrational signals are compared to acoustic signals or sounds (for example L20 and L102). This is incorrect from a physical point of view, sounds and substrate-borne signals are different kind of waves, they have different properties, and they are detected by different sensors and measured with different units. What the authors, probably, wanted to communicate is that the analysis used for sounds can be applied to recorded files of substrate-borne vibrations, but it should be better explained to avoid confusion. Second, the section “Data collection and filtering” must be improved. Too many information is missing (see list below) that are essential to ensure reproducibility.

Overall, the English is relatively correct, but it should be improved before publication (see below a partial list of mistakes).

List of details missing from the “Data collection and filtering” section:

L152-155: Additional information are needed about the measuring procedure. What kind of probe has been used (model, manufacturer)? Was the probe or the input signal calibrated and how? What software has been used for recording? What parameters have been used for acquisition (frequency rate, overlap)?

L155: “relatively quiet environment” is not clear, was the room isolated and how? Was it just a normal room with no people talking? Did the authors used an antivibration table? Please, be more specific in the terminology and description of the methods.

L157-158: What time? How many times per day and per trunk?

L167: How was the background noise removed? (was the file cut or a filter applied?) What software has been used?

L172-173: Please, add the geographic coordinates of the five sites.

Some typo and language mistakes:

L127: “is a” (the n should be deleted)

L133: “boring” (the r is missing)

L147: “have been collected”

L151: “of equivalent length”

L151: “after the selection,”

L153: “those” should be replaced by “the” (twice in this line)

L157: The larvae is plural, I expect “were”

L184: What does “from themselves” mean here?

L205: “came to out sight” is not clear, please clarify

L319: “were recorded with a”

Author Response

The entomological background and implication for pest management of the manuscript is strengthened in revision, as requested by academic editor. The manuscript may be more qualified for Insect with more entomological background. 

I have referenced literature about biotremology and understood the differences of acoustic signals and boring vibrations. All "sound" in manuscript are changed to "vibration".

Details missing from the “Data collection and filtering” section are supplemented. The probe was jointly developed by Beijing Forestry University and Beihang University. Since it was not the major subject of the manuscript, the specification of it may be redundant.

All language mistakes are corrected.

Round 2

Reviewer 1 Report

 The revised MS version addresses the previous comments on the terminology of sound/ vibration, the information on background noise, and the potential use of the obtained information to protect forests from boring insects. The latter two aspects are covered shortly while referenced, but could be specified. The nature of the report is rather technical while the application of this signal analysis method to entomological problems in forest system opens diagnostic possibilities, and overall contributes to the field of applied insect biotremology.

There are some technical aspects which should be explained in more detail in the manuscript. The text would still be improved from a native speaker and below, I point out some issues which should be clarified in phrasing or formatting:

L 15    To deal with this problem, a new method is presented here….

L 28, 59          check for double spaces in the inserted text 

L 51    Among forest pest species, the trunk-boring beetles are particularly difficult to manage. Such species include…

L 60    Are there references where the behaviours and accompanying signals are analysed?

L 62    “…whether the trunk is infected [3].To date…“ – insert space after dot

L 66    The idea of automatically detecting wood boring larvae by their vibrations is not a new one. However, it is constantly interfered by the noise recorded simultaneously.“

The two arguments ar not really linked as suggested by the „However…“. Consider to rephrase e.g.,

"The idea of automatically detecting wood boring larvae has been investigated before, but the signal detection/analysis is constantly hampered by the background noise recorded simultaneously.“

L 92    „Though there are some studies in literature,…“ – Though there are some studies available,…

L 107  vibrations?

L 109  five borer pests: what does this mean, are these individuals, or different species?

L 117  …most existing discrimination methods…

L 174  what was the source of the piezo probe (company)?

L 194  Afterthe – introduce space

L 197  Here, the signals analysed in the study are identified as boring vibrations, but in the introduction, stridulation and percussion are mentioned. If these are anylsed in previous studies, could you emphasise that the present study addresses different vibrations. Or include boring vibrations in the intro as well, characterise it (does it last longer?), and briefly explain the differences?

L 197  How was the probe calibrated?

L 203  “all person left the room” – all individuals left the room

L 429  Axis labelling in Fig. 4 is rather small and gets coarse when enlarged – at least in the file version provided for reviewing. I suggest to increase the font size.

L 459  As stated in a previous study…

L 481  This is probably caused…

L 499  Could you include examples for chemical and biological control measures which could be taken in this case?

L 501  Concluding sentence: something seems to be missing in the sentence: “Consequently, it contributes to…?”

L 573  set species name in italics

Author Response

Our study do not involve vibrational signals produced bystridulation and percussion, they have been deleted from the menuscript.

The probe we utilized was jointly developed by Beijing Forestry University and Beihang University. It is explained in the "Data Collection and Filtering" section.

The probe was calibrated by the following procedure. We used a signal generator to generate sinusoidal wave with frequency between 500 Hz to 10k Hz to simulate boring vibrations. The generated signals were then transmitted to a power amplifier and amplified for a vibration exciter. The probe was mounted to the vibration exciter to pick up vibrations. An oscilloscope displayed the original signal and the signal detected by the probe for comparison. The calibration procedure has been added to the manuscript as commented by the reviewer.

Additional information about noise is added to the third paragraph of Introduction.

Fig. 4 is redrawn with larger font size for axis labels. The resolution of Fig. 4 is 300 dpi as requested in the Instructions for Authors of Insects. The label is still clear when magnified to 400% in Word version of manuscript. 

 Examples for chemical and biological control are added to the second paragraph of Discussion.

The Species name in reference 23 is set to italics. All species names in references are checked.

Language mistakes in comments are corrected accordingly.

Reviewer 2 Report

I read the response of the authors and the revised manuscript, which I consider appropriate for publication in Insects.

Author Response

Thank you for your comments and suggestions in the previous revision.